

# Var3PPred: variant prediction based on 3-D structure and sequence analyses of protein-protein interactions on autoinflammatory diseases

Alper Bülbül[1], Emel Timucin[1], Ahmet Can Timuçin[2], Osman Uğur Sezerman[1] and Eda Tahir Turanli[2]

[1] Department of Biostatistics and Bioinformatics, Institute of Health Sciences, Acibadem University, Istanbul, Turkey

[2] Department of Molecular Biology and Genetics, Faculty of Engineering and Natural Sciences, Acibadem University, Istanbul, Turkey

Corresponding author
Alper Bülbül,
abdullah.bulbul@live.acibadem.edu.tr

## ABSTRACT

We developed a pathogenicity classifier, named Var3PPred, for identifying pathogenic variants in genes associated with autoinflammatory disorders. Our comprehensive approach integrates protein-protein interaction analysis and 3D structural information. Initially, we collected a dataset of 702 missense disease-associated variants from 35 genes linked to systemic autoinflammatory diseases (SAIDs). This dataset, sourced from the Infevers database, served as the training and test sets. We used the SMOTE algorithm to balance the dataset comprising 130 benign and 572 pathogenic variations. Our approach included 3D docking analysis of protein-protein interactions, utilizing data from the STRING and Intact databases. We weighted ZDOCK and SPRINT values in accordance with HGPEC gene rank scores for robustness. Additionally, we integrated sequential and structural features, such as changes in folding free energies ($\Delta\Delta G$), accessible surface area, volume, per residue local distance difference test (pLDDT) scores, and position specific independent count (PSIC) scores. These features, calculated using PyRosetta and AF2 computed structures, provided insights into amino acid conservation at variant positions and the impact of variants on protein structure and stability. Through extensive hyperparameter tuning of six machine learning algorithms, we found the random forest classifier to be the most effective, yielding an AUROC of 99% on the test set. Var3PPred outperformed three other classifiers, SIFT, PolyPhen, and CADD, on an unseen test set of a SAID-related gene. This demonstrates its capacity for pathogenicity classification of SAID variations. The source code for Var3PPred and the predictions for all 420 missense variants of uncertain significance from the Infevers database are available on GitHub: (https://github.com/alperbulbul1/Var3PPred).

## INTRODUCTION

The challenge of discerning the pathogenicity of variants of uncertain significance (VUS) constitutes a significant issue within clinical genomics. Variants that fall into this category are genetic alterations that exhibit subtle or ambiguous functional consequences, rendering their classification as either benign or pathogenic difficult. Systemic autoinflammatory diseases (SAIDs) are marked by recurring bouts of inflammation and fever, which arise due to dysfunction in the inflammasome mechanism, occurring independently of autoantibody reactions or microbial infections. The fact that more than half of patients do not harbor any pathogenic variant in formerly linked disease genes, coupled with the high prevalence of patients carrying VUS, poses a significant challenge in the diagnosis and treatment (*Karacan et al., 2019*). The Infevers database serves as a valuable resource in the field of autoinflammatory disease, which catalogs a vast collection of data related to genetic variations, their associated clinical presentations, and their implications in autoinflammatory disorders (*Van Gijn et al., 2018*).

Numerous endeavors have been undertaken to categorize VUS within the Infevers database, employing a combination of clinical curation and advanced computational techniques such as the random forest algorithm, which achieved an AUROC of 0.91 in the VIPPID model (*Papa et al., 2021*; *Fang et al., 2022*). However, these efforts have yet to yield a comprehensive elucidation or a revelation of the intricate 3D effects of VUS on protein structures, including their potential to disrupt interactions crucial to causative proteins. The 3D structure of proteins is important in variant prediction because it provides crucial information about the protein's function, stability, and interaction with other molecules or proteins. Variants can lead to changes in the 3D structure, affecting the protein's ability to perform its biological functions, which can have implications for disease development and progression (*Caswell et al., 2022*). To facilitate protein-protein docking the ZDOCK algorithm is used in this study. ZDOCK is a widely recognized algorithm that performs a full rigid-body search of docking orientations between two proteins, utilizing a fast Fourier transform (FFT) approach. This allows for the rapid screening of potential docking configurations, making it an invaluable tool in the initial stages of protein complex prediction. Additionally, ZDOCK incorporates a novel pairwise statistical energy potential, which enhances its ability to discriminate between near-native and non-native protein-protein interactions. Consequently, ZDOCK's combination of speed and accuracy makes it a powerful tool in the field of computational biology for predicting the structures of protein complexes, which is crucial for understanding biological functions (*Pierce, Hourai & Weng, 2011*).

Moreover, preceding methods for predicting variant impact, which lack disease-specificity, have traditionally incorporated evolutionary conservation analysis at variation sites. In recent times, these methods have converged with those that assess alterations in the 3D structural stability of proteins. In a recent development, the MutaBind2 tool, an upgraded version introduced by *Zhang et al. (2020)*, employs the $\Delta\Delta G$ values of interacting proteins to forecast variant pathogenicity. However, a limitation of this tool arises from the fact that most proteins lack experimental 3D structures, which consequently precludes

their inclusion in the calculations. Among the widely recognized and commonly employed prediction tools, Sorting Intolerant From Tolerant (SIFT), Polymorphism Phenotyping (PolyPhen), and Combined Annotation Dependent Depletion (CADD) stand out for their robust predictive capabilities, which are described below.

In a seminal article, *Chasman & Adams (2001)* demonstrated that changes in the three-dimensional configuration of proteins hold promise for predicting variant pathogenicity (*Wang & Moult, 2001*). Initial methods that were rooted in evolutionary principles, like Henikoff's approach centered on base changes, were introduced (*Ng & Henikoff, 2001*). Subsequently, more sophisticated methodologies evolved, incorporating techniques for sequence alignment and calculating the impacts of amino acid substitutions, yielding heightened precision in predictions. Among these methodologies, SIFT and PolyPhen-2 have emerged as frequently employed tools for their efficacy (*Vaser et al., 2016*; *Adzhubei et al., 2010*).

Machine learning techniques entered the prediction landscape around 2004 with Cai's Bayesian network learning approach, based on the evolutionary conservation of regions (*Cai et al., 2004*). Meta-prediction methodologies also surfaced, integrating the outcomes of preceding variation prediction methods into a unified pathogenicity assessment. A prime exemplar of this paradigm is CADD, which amalgamates diverse pathogenicity approaches, including GWAS, SIFT, PolyPhen, and more, employing the SVM algorithm to conduct pathogenicity evaluations (*Kircher et al., 2014*).

A recent study has shown that the prediction tools CADD, SIFT, and PolyPhen have achieved ROC AUC values of 0.88, 0.88, and 0.87, demonstrating their high accuracy in variant impact prediction (*Frazer et al., 2021*). We compared the predictive performance of our model with established tools such as SIFT, PolyPhen, and CADD for *MEFV* exon 2 and exon 10 variants. The SIFT model exhibited an accuracy of approximately 74%, with five false positives and 13 false negatives among the 59 variants analyzed. PolyPhen showed a slightly higher accuracy of about 84%, with no false positives but 11 false negatives. The CADD method, which classifies variations with a Phred score above ten as pathogenic, achieved an accuracy of around 75%, with eight false positives and nine false negatives. In contrast, our model outperformed these established methods, demonstrating superior predictive accuracy with eight false positives and, notably, no false negatives.

## METHODS

### Dataset construction

A total of 35 genes pertinent to autoinflammatory diseases were listed along with their respective Online Mendelian Inheritance in Man (OMIM) accession number of associated diseases. A total of 702 missense variations found in these genes were collected from the Infevers database (*Van Gijn et al., 2018*). Interaction partners of these genes were obtained from the STRING and Intact databases using Cytoscape (*Shannon et al., 2003*; *Szklarczyk et al., 2019*; *Hermjakob et al., 2004*). A total of 191 distinct interacting partners of 35 autoinflammatory disease-related proteins were identified. The protein sequences of 226 genes, which include the 35 autoinflammatory disease-related proteins and their
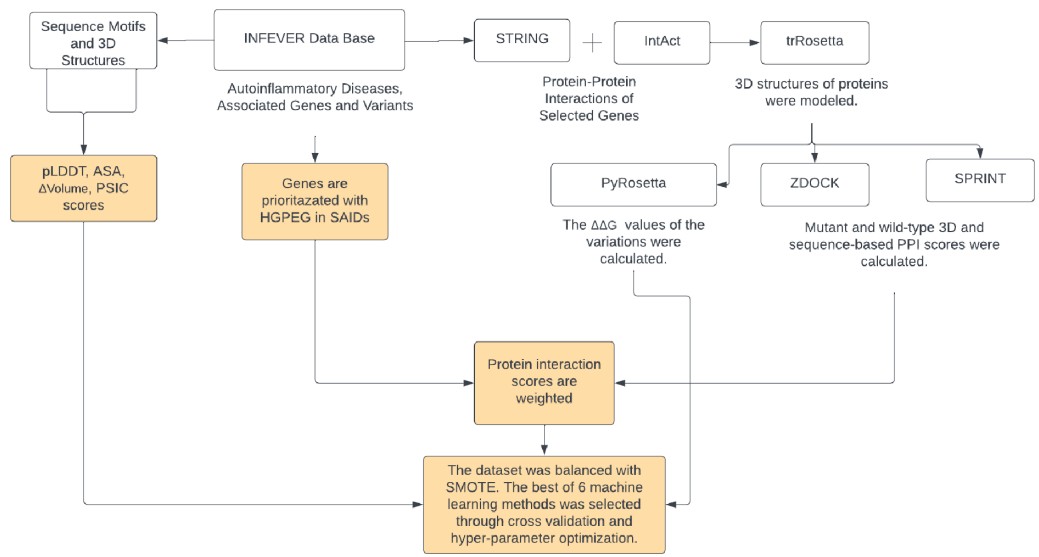

**Figure 1** **The Var3PPred workflow illustrates the systematic data acquisition and processing pipeline.** Variation data is extracted from the Infevers database. Protein-protein interaction data is collated from the STRING and IntAct databases. The depicted steps include the retrieval of sequences from the UniProt database, 3D structural modeling *via* trRosetta, PPI analysis conducted using ZDOCK and SPRINT (values of which are weighted by HGPEC scores), and ΔΔG computations accomplished with PyRosetta. Furthermore, the analysis and evaluation of missense variations involve tools that assess accessible surface area (ASA), pLDDT, and position-specific independent counts (PSIC). The dataset's variability is managed using the SMOTE algorithm, and the predictive model is generated employing the random forest method. Diagram created *via* Lucidchart (https://lucid.app).

191 interacting partners, were retrieved from UniProt (*Consortium, 2019*). 3D structures of these 226 proteins were modeled by trRosetta and AF2 (*Yang et al., 2020*; *Jumper et al., 2021*). Pathogenic and benign variations in the 35 autoinflammatory genes were analyzed for their ZDOCK and SPRINT interaction scores, weighted with 191 interaction partners sourced from the STRING and IntAct databases (*Szklarczyk et al., 2019*; *Hermjakob et al., 2004*). 3D structures of the proteins were used to obtain the protein-protein complexes by ZDOCK. Sequence-based PPI scores were obtained by SPRINT tools. An enhanced ΔΔG score has been computed using the PyRosetta Python package's weighted score function (*Chaudhury, Lyskov & Gray, 2010*), in conjunction with the importance scores of HGPEC genes across 35 autoinflammatory diseases. For the 702 missense variations identified, scores for ASA, PSIC, pLDDT, and volume of variation were compiled. To overcome the imbalance between pathogenic and benign variations, the SMOTE algorithm was used to produce models with higher prediction rates by balancing the ratio between benign and pathogenic variants. The workflow of Var3PPred was presented in Fig. 1.

## Protein structure modelling

The three-dimensional structures of SAID-associated proteins and interaction partners were modeled using trRosetta (*Yang et al., 2020*). The initial step of 3D structure modeling involved performing multiple sequence alignments (MSAs) for each protein sequence utilizing the HHblits tool with the Pfam database (*Remmert et al., 2012*;

*El-Gebali et al., 2019*). These MSAs serve as crucial input for the subsequent trRosetta-based structure prediction, providing evolutionary information that aids in defining the structural constraints.

## Analysis of variations' protein structure and motifs: $\Delta\Delta G$, pLDDT, ASA, and PSIC scores

The process of predicting the protein structure entails a deep residual network-based prediction of inter-residue orientations and distances, which guide the folding of the protein into its final 3D form. Furthermore, to assess the structural implications of 702 variations on the 3D structure of corresponding proteins, they were modified using PyRosetta (*Chaudhury, Lyskov & Gray, 2010*). Also, the change in the Gibbs free energy ($\Delta\Delta G$) due to the variation was calculated with PyRosetta's $\Delta\Delta G$ module.

From the AlphaFold Protein Structure Database, we extracted the AF2 computed structures of 35 proteins. Using these structures, we obtained/calculated the pLDDT scores, which quantify the local protein structure prediction confidence and the relative surface accessible surface area by the DSSP (Dictionary of Secondary Structure of Proteins) module available in the Biopython package (*Cock et al., 2009*). The PSIC scores for both the wild-type and variant amino acids were obtained from the pre-computed profiles of PolyPhen database. These scores provide insights into the possible impacts of amino acid substitutions on protein function and stability.

Handling missing data involves a systematic imputation strategy where each feature exhibiting missing values is modeled as a dependent function of other available features, employing a sequential, round-robin methodology. This process is operationalized using the fancyimputer package in Python, which facilitates advanced statistical techniques for effective and accurate data imputation (*Rubinsteyn & Feldman, 2016*). The imputation strategy employed is IterativeImputation, a multivariate imputer that estimates each feature from all the others. This approach imputes missing values by modeling each feature with missing values as a function of other features in a round-robin fashion.

## 3D and sequence-based protein–protein interaction analysis

Structural protein files in pdb format obtained from the trRosetta tool were preliminary for the ZDOCK tool (*Pierce, Hourai & Weng, 2011*). Structural protein files in pdb format received from the trRosetta tool were preliminary for the ZDOCK tool. Surface amino acids of proteins are marked, and electrostatic charges of atoms are calculated according to UNICHARMM data. The GNU-parallel tool is used to run processes in parallel on high-performance computers (*Tange, 2018*). Interaction partners of SAID-associated genes obtained from STRING and Inact databases were examined with ZDOCK, and the highest score was obtained. Proteins' sequences in fasta format are accepted in the prepared pdb files. Similar sub-sequences are obtained using the PAM120 scoring matrix. Then, interaction scores of mutant interactions are obtained by trained interactions (*Li & Ilie, 2017*). Pairwise interaction scores of Protein interactions ZDOCK and SPRINT are shown in Table S2.

## Weighting interaction scores with the HGPEC gene rank score in SAIDs

Random walk with restart algorithm on a heterogeneous network (RWRH) is used to get the genes' SAIDs disease relatedness score.

$$P^{t+1} = (1-\gamma)W'P^t + \gamma P^0 \tag{1}$$

The RWRH formula is represented at Eq. (1) as *Le & Pham (2017)* described, where $P^t$ is the probability vector at iteration $t$t, $\gamma$ is the restart probability, and $W$ is the transition probability matrix of the network. The matrix $W$ is constructed by combining the gene-disease and disease-gene interaction probabilities, normalized to ensure that each row sums to one. The initial probability vector P0 is set based on the diseases of interest, with non-zero probabilities assigned to the corresponding nodes in the network. The RWRH algorithm iteratively updates the probability vector until convergence, with the final probabilities indicating the relatedness of each gene to the specified diseases. By tuning the parameter $\gamma$, researchers can control the balance between exploration of the network and focus on the initial disease nodes, allowing for the identification of both closely and distantly related genes.

The pre-existing 'Disease Similarity Network 15', integrated within the HGPEC toolset, is a pivotal resource for discerning disease similarity in SAIDs. The prioritization process ranks genes based on their relevance in the recommended human Protein-Protein Interaction (PPI) network and similar selected diseases incorporated in the HGPEC (*Le & Pham, 2017*). These rank scores will later be used in weighting ZDOCK and SPRINT data, as shown in Table S3.

SPRINT and ZDOCK interaction scores were then calculated for all interaction partners of SAID-related genes corresponding to a single mutation, yielding a comprehensive interaction score.

The total score for a mutation was calculated by averaging all weighted interaction scores associated with the mutated protein, as depicted by Eq. (2):

$$W_v = \frac{\sum [G - V_i]Vr_i}{Vr_{sum}}. \tag{2}$$

G is the interaction score of the mutated protein with its interaction partner in the network. $V_i$ is the interaction score of a wild-type protein with its interaction partner in the network. $Vr_i$ is the HGPEC gene rank score of the interaction partner of SAID-related gene. $Vr_{sum}$ is the sum of all $Vr_i$ values for all interactions of the mutated protein.

The overall fraction summarizes all the individual weighted interaction scores for the mutated protein. It is divided by the sum of all HGPEC gene rank scores of interaction partners of SAID-related genes. This fraction provides the average weighted interaction score for the mutated protein across all its interactions.

## Machine learning models training and evaluation

In selecting these six machine learning algorithms for our study, we aimed to encompass a diverse range of approaches to evaluate their performance on complex biological data. Random forest and AdaBoost, both tree-based ensemble methods, were chosen for their

proven effectiveness in handling high-dimensional data and their ability to improve predictive accuracy through ensemble learning. Linear SVM and logistic regression represent linear models, which, despite their simplicity, are powerful tools for classification tasks and provide a baseline for comparison with more complex models. KNN, a non-parametric method, was included for its simplicity and intuitive approach to classification based on proximity in the feature space. Lastly, QDA, a parametric discriminant analysis technique, was selected for its ability to model complex decision boundaries when the data is assumed to follow a Gaussian distribution. Together, these algorithms offer a comprehensive overview of the performance of different machine learning approaches on biological data, with the Python Scikit-learn package providing a consistent and efficient framework for their implementation and evaluation (*Pedregosa et al., 2011*).

Before training models, the imbalance between benign (130) and pathogenic (572) variants number in the dataset was balanced by employing the Synthetic Minority Over-sampling Technique (SMOTE) (*Chawla et al., 2002*).

These models were trained on a dataset of 80% missense variations dataset, with respective $\Delta\Delta G$ values, weighted SPRINT scores, ZDOCK scores, pLDDT scores, ASA, mutant amino-acid volumes, as well as wild-type and mutant PSIC scores. Subsequently, 20% of the dataset was partitioned off for model testing. Each model's hyper-parameters were optimized according to their performance on the training set. The explored hyper-parameters for each model included:

- 'Random forest': Number of estimators (700, 800, 900,1000), maximum depth of the trees (None, 5, 10, 20, 30), minimum samples required to split an internal node (2, 5, 10), and minimum samples required at each leaf node (1, 2, 4).
- 'Linear SVM' and 'Logistic Regression': Regularization parameter (0.1, 1, 10, 100).
- 'K-nearest neighbor': Number of neighbors to use (3, 5, 10, 15).
- 'AdaBoost': Number of estimators (10, 50, 100, 200).

Following training, models were evaluated on the test data using metrics, including Receiver Operating Characteristic Area Under Curve (ROC AUC), F1 score, balanced accuracy, accuracy, and Matthews correlation coefficient (MCC). Moreover, the models were subjected to rigorous validation *via* 20-fold cross-validation on the training data, with performance assessed using the same metrics as applied to the test predictions.

We compared various features and their interaction pairs within the context of the previously mentioned machine learning algorithms. To optimize performance, hyperparameter tuning was conducted for each of these models.

Feature importance, decision-making progress, and features paired interaction impact on model decision of selected models were evaluated with SHapley Additive exPlanations (SHAP) algorithm (*Lundberg & Lee, 2017*).

An in-depth analysis of the features and their interactions was conducted by generating scatter plots for paired features, providing a comprehensive visual representation of the data distribution. To delineate the decision boundaries established by the predictive models, 2D contour plots were produced. Thereby allowing an intuitive understanding of the models' decision-making patterns. These models are also trained in 80% of the balanced dataset.

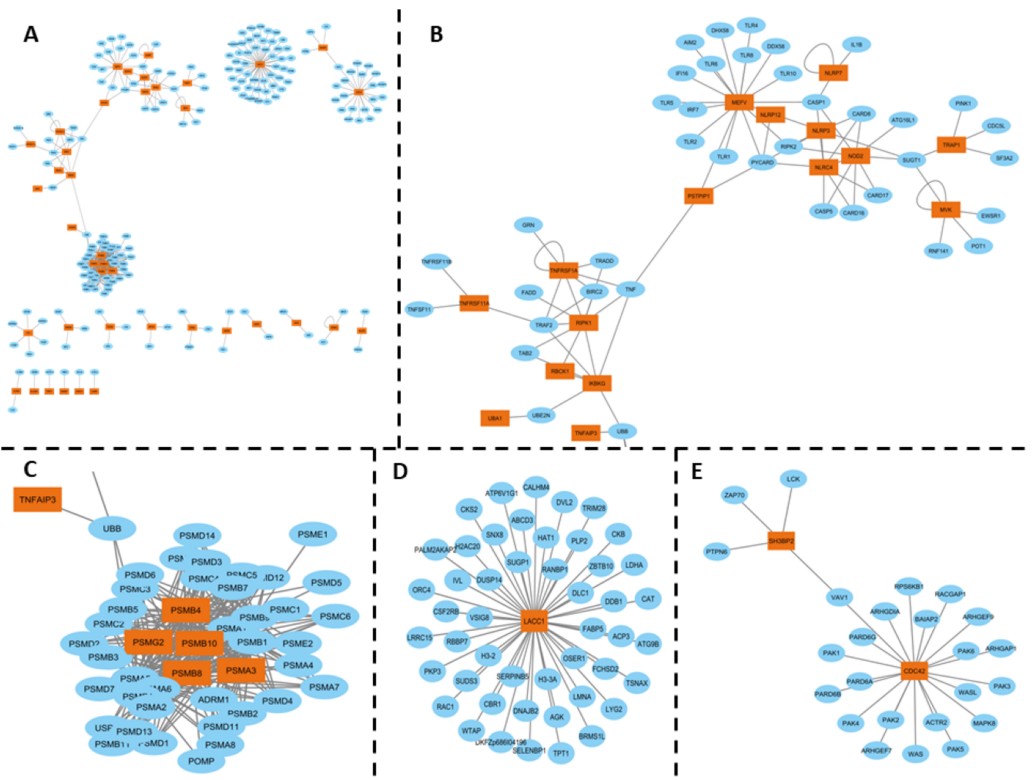

**Figure 2** (A) PPI network of SAID-associated genes and their interaction partners, with orange representing SAID-associated proteins and blue representing their interaction partners. (B) A subnetwork spotlighting the inflammasome complex, encompassing the following genes: *MEFV, PSITPIP1, NLRP2, NLRP3, NLRP7, NLRP12, NLRC4, NOD2, TRAP1, MVK, TNFRSF11A, RIPK1, IKBKG, TNFRSF1A,* and *TNFAIP3*. This complex plays a critical role in inflammatory responses. (C) Immunoproteasome complex participates in Major Histocompatibility Complex (MHC) post-processing. The genes involved include *PSMB4, PSMG2, PSMB10, PSMB8,* and *PSMA3*. (D) *LACC1* gene has been associated with juvenile idiopathic arthritis, and its interaction partners (E) *SH3BP2* and *CDC42* PPI interactions, which associated with Cherubism, a rare autoinflammatory bone disorder, while *CDC42* is associated with the CDC42-associated autoinflammatory disease (CDC42-AID).

The performance of each model was evaluated using the remaining 20% of the data, which was reserved for testing.

## RESULTS

The inflammasome complex subnetwork underscores the critical role of this intracellular protein complex in the inflammatory responses associated with SAIDs. Genetic variants in the genes encoding inflammasome components can lead to autoinflammatory diseases. For example, mutations in the *MEFV* gene, which encodes pyrin, are associated with Familial Mediterranean Fever (FMF), a disease characterized by fever, serositis, and arthritis (*Aksentijevich & Schnappauf, 2021*) (Fig. 2B).

Inflammasomes are cytosolic multiprotein oligomers of the innate immune system responsible for the activation of inflammatory responses. Variations or dysregulation

in these genes could lead to aberrant activation of inflammasomes, thereby triggering the uncontrolled inflammation observed in SAIDs. Moreover, the interconnectedness observed in the inflammasome complex subnetwork suggests that these genes might not function in isolation but contribute collectively to the disease phenotype through synergistic or cumulative effects. It consists of various sensors, including *NLRP1*, *NLRP3*, *NLRC4*, and pyrin, which detect pathogen-associated molecular patterns (PAMPs) and damage-associated molecular patterns (DAMPs). The activation of these sensors leads to the assembly of the inflammasome complex, which in turn activates caspase-1, resulting in the processing and release of pro-inflammatory cytokines IL-1β and IL-18 (*Fernandes et al., 2020*) (Fig. 2B).

The immunoproteasome complex, which includes genes such as *PSMB4*, *PSMG2*, *PSMB8*, *PSMB10*, and *PSMA3*, plays a crucial role in the post-processing of major histocompatibility complex (MHC) proteins. Mutations in these genes have been associated with autoinflammatory diseases, such as proteasome-associated autoinflammatory syndrome (PRAAS) and Chronic Atypical Neutrophilic Dermatosis with Lipodystrophy and Elevated temperature (CANDLE) syndrome. These diseases are characterized by systemic and organ-specific inflammatory damage resulting from defective regulation of the innate immune system. For example, compound heterozygous mutations in the PSMB8 gene have been identified in patients with CANDLE syndrome, leading to severe systemic inflammation, lipodystrophy, and growth retardation (*Boyadzhiev et al., 2019*) (Fig. 2C).

Further highlighting the intricacy of the molecular landscape in SAIDs, interactions involving genes like *LACC1*, associated with Juvenile Idiopathic Arthritis, as well as *SH3BP2* and *CDC42*, associated with distinct autoinflammatory disorders, were also observed (Figs. 2D, 2E). The HGPEC tool is instrumental in dissecting this complexity. This gene prioritization tool considers several crucial factors: disease similarity networks, protein-protein interactions, and established associations between diseases and genes .

We have emphasized the significance of detailing their functional domains, interactions, and the location of selected mutations. Additionally, the domains of pathogenic and benign variations are represented in Table S4, based on InterPro domain predictions. The analysis of variant data revealed a predominant occurrence of certain protein domains impacted by genetic variations: the GHMP kinase N-terminal domain, the TNFR/NGFR cysteine-rich region, and the NACHT nucleoside triphosphatase domain. These domains are frequently represented among the variants studied.The distribution of 703 benign and pathogenic variants is delineated in Fig. S3.

Applying the HGPEC tool in this context has led to identifying and ranking 226 genes associated explicitly with SAIDs. These genes have been systematically listed and ranked based on their relevance to these diseases, as detailed in Table S1.

Upon examining the interaction data, benign interactions were quantified at 1,290, while those involving pathogenic variations stood at 4,249. Intriguingly, the mean ZDOCK score for benign variations surpassed that of pathogenic variations, implying that benign variations foster stronger interactions compared to both wild-type protein interactions and those involving pathogenic variations.

Figure S1 presents pair-wise unweighted density plots of ZDOCK and SPRINT interaction scores for benign variations, wild-type (no-mutation) interactions, and pathogenic variations. The scatter plots show that benign variations exhibit a more diffuse distribution, indicating a wider range of interaction scores. In contrast, the distribution of pathogenic and wild-type variations is more concentrated, with scores falling within a narrower range.

The existence of multiple saddle points in the interaction between pathogenic and benign variants can likely be attributed to the intricate interplay among diverse proteins where these variations manifest multiply. This complexity is underscored by the findings in Table S2, which reveal that specific genes harbor a higher prevalence of well-documented pathogenic variations. Among these, *MEFV*, *NOD2*, *MVK*, *NLRP3*, and *TNFRSF1A* stand out as genes with the most significant associations with diseases.

The observed range of interaction scores for benign proteins extends from 184 to 15,508, indicating a broad spectrum of interaction intensities within non-pathogenic contexts. Contrastingly, interactions involving wild-type proteins exhibit a narrower range of scores, spanning from 28.5 to 851, with a noted variability as indicated by an unspecified standard deviation. This suggests a more constrained range of interaction dynamics for normal protein forms. Additionally, a contributing factor to the elevated scores observed in the SPRINT ranking system can be attributed to the high HGPEC rank scores associated with proteins that possess pathogenic variations. This correlation implies that proteins with disease-causing mutations tend to have higher interaction and rank scores in these specific assessment systems, highlighting the distinct functional implications of pathogenic *versus* benign protein interactions.

Variations localized in such less accessible regions are believed to have a heightened potential to disrupt the protein's structural and functional dynamics, contributing to disease manifestation (*Laddach, Ng & Fraternali, 2021*). Conversely, the higher ASA in benign variations suggests these variations make more exposed regions or don't change the ASA in the wild-type, where they might not significantly impact the protein's function. ASA showed a mean of 0.498924 for benign and 0.216563 for pathogenic variations.

The volume of mutants possibly showed a mean of 0.496082 for benign variations and 8.850164 for pathogenic variations (Fig. 3). The more considerable average change in residue volume in pathogenic variations might suggest that these variations could significantly alter the protein's structure in a specific domain.

This structural alteration could impact the protein's function, possibly explaining the pathogenic nature of these variations. Conversely, the more minor change in volume in benign variations might indicate less structural disruption, allowing the protein to maintain its function. The mean pLDDT score, indicative of structural confidence, was 75.115158 for benign variations and 91.201447 for pathogenic variations. Higher pLDDT scores in pathogenic variations might suggest that these variations are located in structurally confident regions of the protein. A high pLDDT score is indirectly related to the robust structure of domains, meaning highly confident regions are more rigid according to the high correlation of experimental structural models. Variations in those regions may cause unaccepted malformation in protein function followed by protein structure. In contrast,
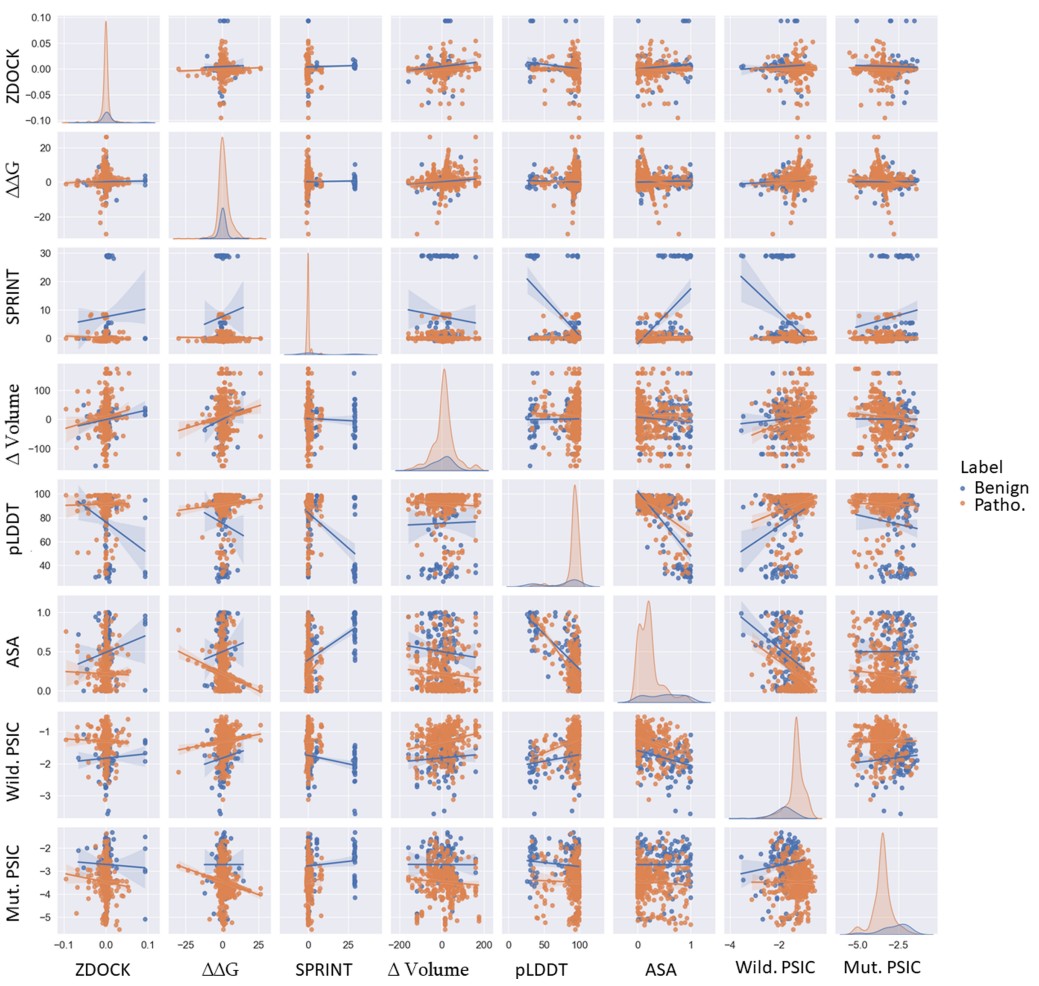

**Figure 3** **Pairwise scatter, density, and regression plots of ZDOCK, ΔΔG, SPRINT, ASA, pLDDT, Δ volume, wild-type, and mutant PSIC.** Orange represents benign variants, while blue represents pathogenic variants.

low-confident regions could be considered as more fluctuating domains. These regions could also be pathogenic and benign because loop formation is essential in various protein functions and could be non-conserved regions. Changes in this protein region may not play an important role in protein stability, structure, and function.

PSIC scores for wild-type and mutant proteins showed mean values of −1.83 (benign) and −1.31 (pathogenic) for wild-type PSIC and −2.73 (benign) and −3.49 (pathogenic) for mutant PSIC. Higher wild-type PSIC scores in pathogenic variations might suggest that these variations occur at more conserved positions, where variations could significantly disrupt protein function.

## Comparison of variants features

Examined ZDOCK, SPRINT, ΔΔG, ASA, and volume change of variations; particularly, benign variations show higher means and standard deviations for ZDOCK and SPRINT

values than pathogenic variations. Interestingly, the scenario is reversed for $\Delta\Delta G$ values, where pathogenic variations exhibit higher distribution and mean values than benign ones. This differential pattern can be visualized effectively *via* scatter plots, as shown in Fig. 3. Specifically, pathogenic variations present a broader $\Delta\Delta G$ distribution on the $y$-axis, whereas benign variations show a more concentrated distribution. Conversely, the ZDOCK data offer a wider range for benign variations. Moreover, the pLDDT score shows comparable separability to SPRINT in distinguishing between benign and pathogenic variants. Additionally, the wild-type PSIC and mutant PSIC scores, indicative of conservation values, are negatively correlated, further emphasizing their relevance in variant segregation.

It is also worth mentioning that pathogenic variants show broader separation in the ASA values. Although the mean Volume of variations is similar for benign and pathogenic variants, this feature demonstrates better separation of variants in combination with the pLDDT value. This suggests the potential utility of incorporating these diverse features in comprehensive variant classification frameworks. ZDOCK and SPRINT have a positive correlation of 0.089. This might indicate that variations have a more substantial impact on protein-protein interactions and also tend to have a stronger impact on domain-domain interactions. $\Delta\Delta G$ and the Volume of the mutation have a positive correlation of 0.14, suggesting that variations causing more extensive changes in residue volume might also cause more significant changes in protein stability. SPRINT and pLDDT have a strong negative correlation of $-0.54$, suggesting that variations affecting domain-domain interactions tend to occur in regions of lower structural confidence. ASA and pLDDT have a strong negative correlation of $-0.71$, indicating that residues with higher structural confidence tend to be less exposed. ASA and SPRINT have a positive correlation of 0.462608, suggesting that variations in more exposed residues might have a stronger impact on domain-domain interactions. Wild-type PSIC and ASA have a strong negative correlation of $-0.49$, suggesting that more conserved positions tend to be less exposed. According to Fig. 3, the most separable pair-wised features are pLDDT and volume of mutation, and SPRINT features separate benign and pathogenic variants when paired with all other features.

## Machine learning models

Comparison results with other models are given in Table 1. The best model is the random forest model algorithm according to ROC AUC and balanced accuracy metrics. The comparative analysis of machine learning algorithms for predicting SAIDs associated variations reveals significant variations in the performance of all the tested models. Tree-based methods such as random forest and AdaBoost displayed superior predictive power across multiple evaluation metrics, particularly regarding the ROC AUC, accuracy, and F1 score, achieving scores of approximately 0.995, 0.978, and 0.95 for both models, respectively. Random Forest, an ensemble learning method that operates by constructing multiple decision trees, demonstrated the highest MCC of 0.90, indicating its robust reliability and strong binary classification abilities. Similarly, the AdaBoost algorithm,

**Table 1  Performance metrics for machine learning models on the test dataset.** Each model was evaluated based on ROC AUC, accuracy, F1 score, precision, recall, balanced accuracy, and MCC. The bolded values indicate the highest scores in each metric.

| Models | ROC AUC | Accuracy | F1 Test | Precision | Recall | Balanced accuracy | MCC |
|---|---|---|---|---|---|---|---|
| Random forest | **0.994911** | **0.95** | **0.95** | **0.97** | 0.94 | **0.952555** | **0.904074** |
| Linear SVM | 0.899831 | 0.88 | 0.87 | 0.87 | 0.87 | 0.877333 | 0.754667 |
| Logistic regression | 0.900673 | 0.88 | 0.89 | 0.87 | 0.9 | 0.876339 | 0.754644 |
| K-nearest neighbour | 0.916666 | 0.88 | 0.87 | 0.98 | 0.79 | 0.883302 | 0.774468 |
| Quadratic discriminant analysis | 0.914753 | 0.75 | 0.79 | 0.7 | 0.91 | 0.736952 | 0.509542 |
| AdaBoost | 0.977655 | **0.95** | **0.95** | 0.95 | **0.95** | 0.947428 | 0.894857 |

known for focusing on classification problems where instances are weighted, also presented a high performance with an MCC of around 0.89.

Meanwhile, other models, including linear support vector machine (SVM), logistic regression, K-nearest neighbour, and quadratic discriminant analysis, yielded relatively lower performance metrics, with the ROC AUC ranging from 0.90 to 0.92 and MCC from 0.51 to 0.77. These differences underline the effectiveness of ensemble tree-based methods in handling the complexity and high dimensionality of biological data, yielding promising results for predicting variants. The random forest algorithm has been selected for subsequent investigations based on the performance metrics across multiple machine learning models.

The superior performance of tree-based methods such as random forest and AdaBoost in our study can be attributed to their inherent ability to handle the complexity and high dimensionality of biological data. These methods effectively capture non-linear relationships and interactions between features, which are common in biological datasets. The ensemble approach of random forest, which combines multiple decision trees, helps in reducing overfitting and improving the generalization of the model. Each tree in the ensemble is built on a random subset of the data and features, leading to diverse trees that capture different aspects of the data. This diversity helps in achieving high accuracy and robustness in the predictions.

AdaBoost, on the other hand, focuses on instances that are difficult to classify, iteratively adjusting the weights of these instances to improve the model's performance. This adaptive weighting scheme allows AdaBoost to concentrate on the areas of the data where the model's performance is weak, leading to a more refined and accurate classification (*Tang, Henderson & Gardner, 2021*).

In contrast, other models like linear SVM, logistic regression, K-nearest neighbor, and quadratic discriminant analysis may not be as effective in capturing the complex patterns present in biological data. These models have limitations in modeling non-linear relationships and may struggle with the high dimensionality of the data, leading to lower performance metrics compared to tree-based methods. The linear nature of SVM and logistic regression, the distance-based approach of KNN, and the assumption of a Gaussian distribution in QDA may not be well-suited for the intricate structure of biological data, highlighting the advantage of using ensemble tree-based methods for such tasks (*Singh, 2019*; *Dreiseitl et al., 2001*; *Kim et al., 2011*).

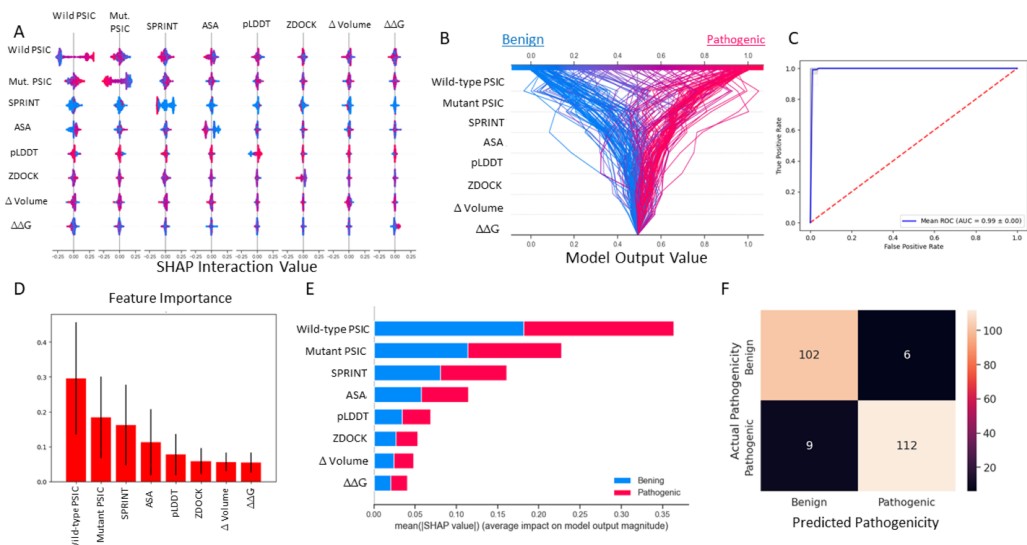

**Figure 4** **Random forest (SMOTE) model metrics.** (A) Random forest model's feature interaction plot according to SHAP values. (B) Random forest model's test data decision plot. (C) Random forest cross-validated (kfold = 20) ROC curve. (D) Random forest model's feature importance bar plot. (E) Random forest model's features impact score according to SHAP value. (F) Random forest model's confusion matrix on test data.

In addition to the models mentioned, we also evaluated a basic shallow fully connected neural network (multilayer perceptron), which achieved an accuracy of approximately 0.85. While this performance is commendable, it still falls short of the 0.98 ROC AUC achieved by the random forest model. Furthermore, a single decision tree model exhibited lower accuracy compared to the random forest, with its performance being sensitive to the random seed used during model creation, indicating variability in its predictive capability.

The most important figures are the wild-type and mutant PSIC scores. The weighted SPRINT interaction score is the third most important feature in the random forest model. Also SHAP impact score of the feature correlated with feature importance (Figs. 4D, 4E). However, in the decision plot, which was employed with 20% test data, some of the test variant's prediction scores were determined with other features like ASA, pLDDT, and ZDOCK (Fig. 4B).

According to the results of this model, the ROC AUC value is 99%, according to the average of 20 split cross-validations. Other metrics are given below.

- F1 test result: 0.990280
- Precision: 0.993148
- Recall: 0.987684
- Accuracy: 0.956031
- ROC AUC: 0.9946581
- Balanced Accuracy: 0.990332
- MCC: 0.981044

In a comprehensive evaluation across six machine learning models, the pairing of SPRINT and wild-type PSIC stood out, achieving an accuracy of 72% in random forest, linear SVM, KNN, and AdaBoost. This combination also exhibited 71% and 68% accuracy in logistic regression and QDA, respectively. Other notable pairings include pLDDT with wild-type PSIC (71% accuracy) and pLDDT with SPRINT (70% accuracy). Interestingly, the ZDOCK and SPRINT combination was examined in two scenarios, registering accuracies of 73% and 71%. Additionally, combinations like wild-type PSIC with ASA, pLDDT with ASA, $\Delta\Delta$G with SPRINT, and SPRINT with ASA achieved accuracies hovering around 67% to 70%. The least-performing pairing was ZDOCK with ASA, which reached 61%. The results emphasize the importance of feature pairing in enhancing predictive accuracy in machine learning models (Fig. 5). The learning curve analysis reveals convergence between test and training data, suggesting proper model fitting Fig. S2.

The train and prediction regions, according to the dual selections of the features of the created models, are shown in Fig. 5. Here, blue regions are determined to predict benign variations, and red areas are determined to predict pathogenic variations. The locations of the dual feature data in these regions are seen as a scatter plot.

## Comparison with other prediction methods (SIFT, Polyphen, CADD)

A comparative analysis was conducted to benchmark the predictive capabilities of our model against three widely used variation prediction methods: SIFT, PolyPhen, and CADD. The test set for this comparison included 59 variations from exons 2 and 10 of the *MEFV* gene.

Our analysis revealed that our model outperforms these established methods in its predictive capacity. The SIFT model yielded a higher number of false positives (five) and false negatives (13), thereby suggesting a lower accuracy. Similarly, the PolyPhen tool, while showing no false positives, had 11 false negatives. The CADD method, which classified variations with a Phred score above ten as pathogenic, resulted in eight false positives and nine false negatives. In stark contrast, our model demonstrated superior performance with eight false positives and, notably, no false negatives. This result underscores the robustness of our model, highlighting its potential to offer improved accuracy for predicting the pathogenicity of genetic variants in the context of SAIDs Fig. 6. Moreover, Var3PPred achieves a higher ROC AUC of 0.99 (Fig. 4C) compared to 0.92 for AlphaMissense, 0.92 for BayesDel, 0.91 for MetaLR, 0.90 for MetaSVM, 0.85 for DEOGEN2, 0.85 for ClinPred, 0.80 for DANN, 0.79 for FATHMM, and 0.75 for Eigen-raw (Fig. 6E).

The prediction results for 420 missense VUS in genes associated with SAIDs and associated domains are available on our GitHub page for further reference and analysis. Our prediction model evaluated VUS, 23.61%) as benign and 76.39% as pathogenic.

## CONCLUSION

Classifying variants of uncertain significance (VUS) can be challenging due to the lack of protein function information of many of these variants. Attempts have been made to classify VUS in the Infevers database using clinical curation and random forest algorithms (*Accetturo et al., 2020*).

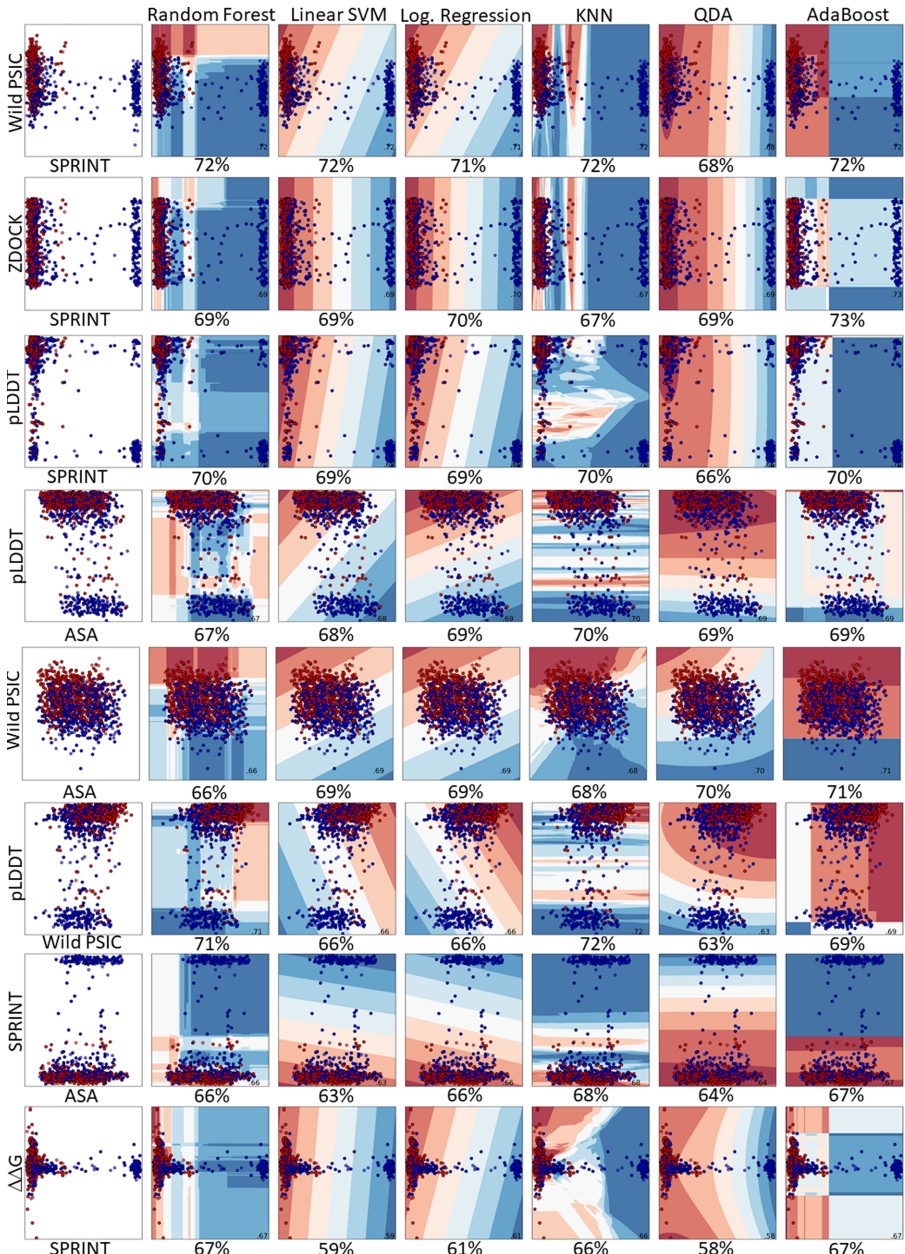

**Figure 5** Pairwise comparison of the feature distributions were shown alongwith the decision boundaries of six hyper-parameter optimized classifiers including random forest, linear support vector machine, logistic regression, K-nearest neighbors, quadratic discriminant analysis and AdaBoost.

Examples of VUS classification in the Infevers database for specific genetic variants associated with autoinflammatory diseases include the V198M variant in the *MVK* gene and the *PSTPIP1* p.Gln219His variant (*Vuran & Berdeli, 2022*). The *PSTPIP1* p.Gln219His variant is not classified in Infevers and is considered a VUS with the ACMG criteria (*Prados-Castaño et al., 2022*).

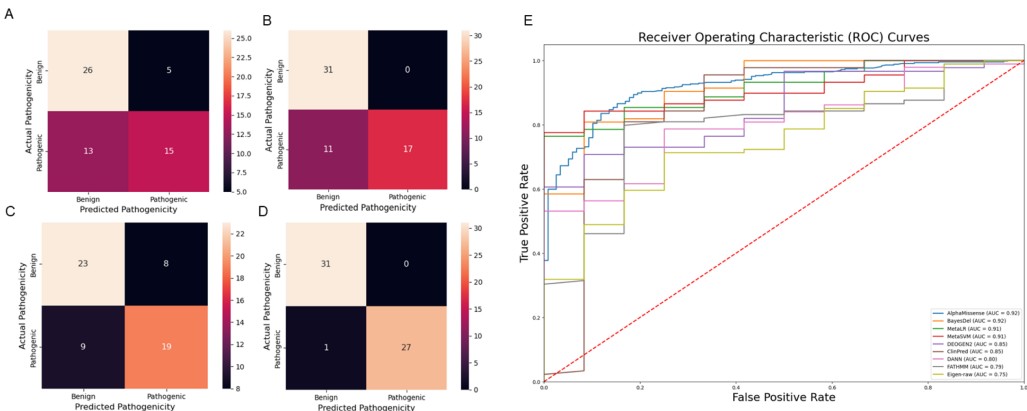

**Figure 6** Comparison of prediction tools with Var3PPred at MEFV exon 2 and 10 variants and overall SAIDs associated variants (A) SIFT prediction tool confusion matrix, (B) PolyPhen prediction tool confusion matrix, (C) CADD prediction tool confusion matrix, (D) Var3PPred confusion matrix, (E) ROC curves were generated for SAID-associated variants using the following prediction tools: AlphaMissense, BayesDel, ClinPred, DANN, DEOGEN2, Eigen-raw, FATHMM, MetaLR, and MetaSVM.

Incorporating protein structural analysis into variant classification workflows can aid in the classification of VUS and provide insights into the mechanisms of pathogenicity. However, the interpretation of variants that cause gain-of-function can be challenging, as their effects on protein structure are likely to be more variable and subtler and may require retention of the active form and structure of a protein (*Caswell et al., 2022*).

This method, developed for predicting variations, is based on protein-protein interaction and the effect of variations on 3-dimensional protein stability to predict variation pathogenicity. Protein-protein interactions affect many intracellular and extracellular functions. These interactions are based on because the impact of variations on protein-protein interactions will disrupt wild-type cell organization (*Titeca et al., 2019*). Previous studies have shown that variations in the only known genes of these diseases cause the disease in most patients. Suppose patients with this disease do not have a known variation in the disease gene. In that case, it is expected to originate from the genes that interact with this gene or from the variant of unknown significant larvae within the gene (*Wong et al., 2020*). In complex diseases, determining the genes according to the subgroups of the diseases is likely to give more meaningful results. For example, gene weighting will provide more meaningful results than the sub-clinical types of multiple sclerosis, relapse remitting multiple sclerosis, secondary progress multiple sclerosis, and primary progress multiple sclerosis (*Kiselev et al., 2019*).

Before using this prediction tool genome-wide, the clinical data of the patients should be examined well. Unbiased test results will not give meaningful results, especially in rare diseases, since the prevalence of the diseases is 1/2000 (*Joseph, Gyorkos & Coupal, 1995*).

In the machine-learning methods used, random forest is the method that gives the best results in both data sets. The ROC AUC value is 99% in the test dataset. Similar patterns were seen in other machine-learning techniques. The KNN model has more false negative values than others. The most important limitation is that only missense variations are used.

In future models, it is aimed to carry out patient-specific variant prediction with variant classification, which can be obtained by using the weighting method according to the symptom similarity of the diseases.

### Funding
This work was supported by the National Center for High Performance Computing (UHeM) (Grant No: 4009452021). The funders had no role in study design, data collection and analysis, decision to publish, or preparation of the manuscript.

### Grant Disclosures
The following grant information was disclosed by the authors:
National Center for High Performance Computing (UHeM): 4009452021.

### Competing Interests
The authors declare there are no competing interests.

### Author Contributions
- Alper Bülbül conceived and designed the experiments, performed the experiments, analyzed the data, prepared figures and/or tables, authored or reviewed drafts of the article, and approved the final draft.
- Emel Timucin performed the experiments, analyzed the data, authored or reviewed drafts of the article, and approved the final draft.
- Ahmet Can Timuçin analyzed the data, authored or reviewed drafts of the article, and approved the final draft.
- Osman Uğur Sezerman analyzed the data, authored or reviewed drafts of the article, and approved the final draft.
- Eda Tahir Turanli conceived and designed the experiments, analyzed the data, authored or reviewed drafts of the article, supervised in genetic perspective, and approved the final draft.

### Data Availability
The data is available in GitHub and Zenodo:
- https://github.com/alperbulbul1/Var3PPred
- Alper Bülbül. (2024). alperbulbul1/Var3PPred: v0.2 (v0.2). Zenodo. https://doi.org/10.5281/zenodo.10675784.

### Supplemental Information
Supplemental information for this article can be found online at http://dx.doi.org/10.7717/peerj.17297#supplemental-information.

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
