# Peer review of "Var3PPred: variant prediction based on 3-D structure and sequence analyses of protein-protein interactions on autoinflammatory diseases"

_PeerJ, doi:10.7717/peerj.17297_

## Round 0.1 · original submission · Major Revisions

Dear Authors,

Reviewers suggested major revisions for your manuscript. Please address all issues raised by the reviewers.

Sincerely.

**Language Note:** The review process has identified that the English language must be improved. PeerJ can provide language editing services - please contact us at [email protected] for pricing (be sure to provide your manuscript number and title). Alternatively, you should make your own arrangements to improve the language quality and provide details in your response letter. – PeerJ Staff

·

Basic reporting

I would like to thank the authors for clear and professional English language that they have used throughout the manuscript. However, there are a couple of issues (given below) that need to be checked:
a. In figure captions, the word “respectfully” was used throughout the manuscript to enumerate two or more items.
b. The meaning of the abbreviation PSIC was written differently in two places: line 25 and the caption of Fig. 1.
c. The provided GitHub link needs to be updated. Unfortunately, it was not accessible.
d. Citation for figures and reference formats need to be adjusted according to the PeerJ standards.
e. In the last column of the Supplementary Table 3, all labels are written as “bening” instead of “benign”.
f. Fig. 6 was not cited in the main manuscript. The x-axis label, “Predicted Patogenicity”, need to be corrected.

Experimental design

Var3PPred workflow incorporates protein-protein interaction data, sequence and structure information and uses these features in six different machine learning (ML) methods, which include rigorous validation, to classify pathogenicity of Systemic Autoinflammatory Disease (SAID) variants. Protein-protein complex structures are one of the important inputs for the Var3PPred tool. The authors stated that ZDOCK was used to generate protein-protein complex structures and the highest docking score obtained from ZDOCK was used as the input for the machine learning part of the workflow. Although the ZDOCK is currently one of the best docking tools, which performs rigid-body docking, the authors should explain the reason for the usage of ZDOCK in their protocol in detail. Moreover, that would be nice to see the results when another rigid-body docking tool such as HADDOCK, ClusPro, or Swarmdock that provides flexible docking. Would the new predictions be very similar to the previous one?

Validity of the findings

Their findings show the capacity of the Var3PPred for classification of benign and pathogenic variants of SAIDs. However, the comparison of their findings to that of AlphaMissense and BayesDel methods would be also helpful to better understand the capacity of Var3PPred.

Reviewer 2 ·

Basic reporting

This paper developed a classifier to identify pathogenic variants in genes associated with autoinflammatory disorders. The dataset consists of 702 variants from 35 genes. Overall, it is clear and unambiguous.

1. In the introduction part, please include some quantitative results for current fields of classification VUS. For example, in line 46-56, what is the accuracy using random forest? Besides RF, what else? If there is no comprehensive by integrating other data such as 3D structure, why? What are the quantitative results of the recent SIFT, PolyPhen, CADD methodologies? Please elaborate on it since I failed to find it in the last 2 paragraphs in introduction part.

Experimental design

The overall experimental and methodology are designed well. Only a few need to be revised:
1. Please reveal the imputation details for line 112 - 116: What package is called? What statistical techniques are used to impute the missing data with what logics?
2. What is the logic to select the 6 models in line 162? Did you try basic shallow fully connected neural network? What is the performance compared to conventional ML models here?

Validity of the findings

The results are well evaluated but some revisions are required.

1. Which paragraphs did the authors refer to the figure 2? What results and insights can be generated from figure 2?
2. The authors only listed the comparison of different metrics across different models, but there is no discussions on the insights about why some models show a better prediction, and why some can not.

Reviewer 3 ·

Basic reporting

The authors address a rather topical issue in bioinformatics and genomics. indeed, the classification of variants of uncertain significance is still one of the major hurdles to be faced in the study of complex diseases. In this study, they develop a new classifier that takes into account information on the 3D structure of proteins and protein-protein interactions for the study of systemic autoinflammatory diseases. In the introduction, the problem is framed in a general way, but it is not sufficiently explored why the authors decide to use 3D protein structure information and protein-protein interactions to develop their classifier. What is needed is some information about the structure of the proteins under study, their functional domains and how they interact with each other, and finally the location of selected mutations within these domains.
The language is not always clear (e.g. lines 278-272, probably the numbers are a typo), there are some repeated sentences, I would suggest a comprehensive review of the English throughout the manuscript, with special attention to the discussion. There is some misspelling and some imprecision of language also in the material on GitHub and in the figure captions.
The supplementary figure 1 should be described more accurately. It would also be interesting to know the variants represented in figures 3 and 5, at least for the most significant cases.
The raw data are partially provided but are not easy for a general audience to read.
On the other hand, such a detailed description of the HGPEC rank score from line 128 seems excessive.

Experimental design

In the description of the initial dataset, it is not clear whether the 191 interacting partners are used or whether they coincide with the 226 genes extracted from UniProt (lines 80-81). The initial dataset should be described more accurately, also specifying the date of data extraction.
The classifier would appear to be overfitting, probably due to a high similarity between the training and test set. There is a risk that the method of imputation of missing data could have contributed. It would be interesting if the authors would elaborate on the reasons for choosing this approach towards missing data over other possible approaches, better if supported by comparison data.

Validity of the findings

It is not clear from the results how much the classifier actually improves on current knowledge. The performance of the classifier on already labelled variants is described, for which there also seems to be an overfitting problem (figure 4C), but then in fact the classification of VUS, reported only on GitHub, does not seem to be commented on anywhere in the manuscript. I cannot find a description of the "Prediction(0-1)" column in the file on GitHub that would help to understand how the authors arrive at the final prediction. One would have to comment better on these results and relate them to what is already known about each VUS, including its localisation with respect to protein domains and the 3D structure of the protein. In this way, the classifier could be better evaluated, understanding whether the data on 3D protein structure and protein-protein interactions significantly improve the current classification. Conclusions should then comment on these results.

---

## Round 0.2 · accepted · Accept

Congratulations your manuscript is found to be acceptable by the reviewers.
Please follow directions of the PeerJ staff for timely publication of your paper.

·

Basic reporting

No comment.

Experimental design

No comment.

Validity of the findings

No comment.

Additional comments

I would like to thank all authors for their effort and time to prepare the revised manuscript.

Reviewer 2 ·

Basic reporting

The revised manuscript has a clear and unambiguous professional English and structures.

Experimental design

The experimental design entails sufficient details in this revised version.

Validity of the findings

The findings are valid.

Reviewer 3 ·

Basic reporting

Overall, the authors address all the required points. There are still some inaccuracies in the test (see e.g. lines 450-451 and 467 and some table headings in the file on GitHub).

Experimental design

The authors describe the method of imputation of missing data used in their study, but do not comment on the reasons why they decided to use this method over others, nor were described any studies comparing different imputation methods. This is certainly a long and laborious task that is probably beyond the scope of this study. However, it would be useful to add a comment on the advantages and limitations of this type of approach and the risks involved in imputing missing data, thus emphasising the consequent limitations this type of approach may have caused to the predictions obtained downstream of the dataset construction.
The other required points were addressed.

Validity of the findings

Overall, the authors address all the required points.